# DNA Damage and Repair in Pancreatic Cancer—The Latest Findings

**DOI:** 10.3390/ijms262010106

**Published:** 2025-10-17

**Authors:** Małgorzata Kozłowska, Michał Mik, Michał Nowicki, Agnieszka Śliwińska

**Affiliations:** 1Department of Nucleic Acids Biochemistry, Medical University of Lodz, 251 Pomorska St., 92-213 Lodz, Poland; malgorzata.kozlowska@umed.lodz.pl; 2Department of General and Colorectal Surgery, Medical University of Lodz, 113 Stefana Zeromskiego St., 90-549 Lodz, Poland; michal.mik@umed.lodz.pl (M.M.); michal.edward.nowicki@umed.lodz.pl (M.N.)

**Keywords:** pancreatic cancer, DNA damage, DNA repair

## Abstract

Pancreatic cancer is one of the most common cancers of the gastrointestinal tract, alongside stomach and colon cancers, yet remains among the least studied. Due to its non-specific symptoms, late diagnosis, and limited treatment options, it is associated with a poor prognosis and high mortality. Major risk factors for pancreatic cancer include smoking, alcohol consumption, pancreatitis, obesity, and type 2 diabetes. These environmental factors can damage DNA through various mechanisms and, if not properly repaired, may initiate carcinogenesis. DNA repair is one of the key mechanisms in cancer prevention. It has been suggested that impaired DNA repair may contribute to the development and progression of pancreatic cancer. The aim of this review is to highlight the link between environmental factors, DNA damage and DNA repair in pancreatic cancer. Environmental exposures can trigger a cascade of molecular events, including ROS (reactive oxygen species) overproduction, oxidative stress, insulin resistance, hyperglycemia, and inflammation, that lead to DNA damage. Additionally, up to 25% of patients with PDAC (pancreatic ductal adenocarcinoma) carry mutations in DDR (DNA damage response) genes, and only 5% of all cases are hereditary. Therefore, increased DNA damage combined with disturbances in DDR creates a condition accelerating pancreatic cancer progression. Further research should focus on DDR pathways as potential targets for screening and therapy. Such an approach could significantly improve early diagnosis and treatment outcomes. Moreover, uncovering the mechanisms linking pancreatic cancer aggressiveness with DNA repair deficiencies may lead to the development of specific biomarkers, enabling early detection and potentially improving patient survival.

## 1. Introduction

The pancreas is an essential digestive and endocrine organ responsible for maintaining metabolic homeostasis [1,2]. Its endocrine function involves regulating blood glucose levels through the interplay of hormones such as insulin and glucagon. The exocrine function supports digestion by secreting enzyme including chymotrypsin for proteins, amylase for polysaccharides, and pancreatic lipase for fats. Pancreatic dysfunction can lead to several diseases, including pancreatitis, diabetes, and cancer. Pancreatic cancers are classified into distinct types based on their clinical features and cellular origin. The first group comprises ductal cancers, the most prevalent of which is PDAC, accounting for approximately 85% of all pancreatic malignancies [3]. This group also includes adenosquamous cell carcinoma, representing 1–4% of malignant neoplasms of the exocrine pancreas, and osteoclastic giant cell carcinoma, which only amounts to 1% of invasive pancreatic cancer [4,5]. The second group includes non-ductal cancers, such as PanNET (Pancreatic Neuroendocrine Tumor), which represent 15%. This group also includes acinar cell carcinoma, pancreaticoblastic tumors, and solid pseudopapillary neoplasms [3]. Due to their non-specific symptoms and the lack of sensitive and effective diagnostic methods in early stages, they are associated with a poor prognosis. It should be noted that the molecular mechanisms of the development and progression of these cancers are poorly recognized. Among the dysregulated cellular processes implicated in their pathogenesis and treatment resistance are cell cycle control, apoptosis, mitochondrial respiration, and DNA repair. The latter is particularly important, as it plays a central role in maintaining genome stability and integrity, key for proper functioning of the pancreatic cells. It is well established that DNA damage is a trigger in the process of carcinogenesis, whereas disturbances in the DNA repair process are associated with cancer progression and resistance to treatment. Therefore, this review aims to present the current state of the art about DNA damage and repair in the pathogenesis of pancreatic cancer.

## 2. Materials and Methods

To summarize the current state of knowledge regarding the role of DNA damage and repair in pancreatic cancer, PUBMED and Google Scholar databases were searched. The following keyword combinations were used: pancreatic cancer, PDAC, PDAC epidemiology, PDAC treatment, DDR, DNA damage, and DNA repair disorders in PDAC. In vitro, in vivo, and clinical studies were reviewed.

## 3. Pancreatic Cancer

### 3.1. The Incidence of Pancreatic Cancer

Globally, the incidence of pancreatic cancer is steadily increasing. According to the World Cancer Research Fund International, pancreatic cancer is the 12th most common cancer worldwide. In Europe and North America, the incidence rates of pancreatic cancer are the highest of all continents and, as maintained by available data, are 8.5/100,000 for North America and 8/100,000 for Europe, respectively, as shown in Table 1.

Therefore, one can see that continents with a high development rate and a greater degree of urbanization have a higher incidence of PDAC compared to developing continents [6]. However, in some African countries, access to adequate medical care may be limited, resulting in lower rates of detection and treatment of the disease [7,8].

### 3.2. Pancreatic Carcinogenesis

Carcinogenesis is a process involving both genetic and epigenetic alterations that, through genomic instability and inflammation, lead to the development of cancer. At the single-cell level, DNA damage can arise due to errors in DNA replication or repair, as well as from chromosomal or epigenetic modifications. Although environmental factors such as smoking, alcohol consumption, obesity and type 2 diabetes play a major role in the development of pancreatic cancer, the underlying molecular mechanisms remain complex. The influence of environmental factors on pancreatic carcinogenesis will be discussed in the next chapter.

Carcinogenesis comprises three stages: initiation, promotion, and progression. The initiation stage is marked by the occurrence of an oncogenic mutation in a single somatic cell, often arising from spontaneous DNA modification. Under physiological conditions, cells possess multiple DNA repair mechanisms that counteract potentially harmful alterations, including mutations. However, due to environmental factors or deficiencies in DNA repair, cells with damaged DNA survive and transform into cancerous cells [9]. DNA damage, if not sufficiently repaired, plays a key role in the initiation phase. Although inflammation is not the central process in this stage, it contributes by generating ROS.

The promotion phase involves the selective clonal expansion of cells that have acquired mutations [10]. During this stage, permutated cells undergo further proliferation. Inflammation plays a dominant role by promoting cell division, increasing the synthesis of pro-inflammatory cytokines and ROS, and modifying the tumor microenvironment. At this stage, mutant cells evade apoptosis, accumulate further mutations that confer immortality, and proliferate in a manner that drives progression. Clonal expansion and genomic instability further impair cellular signaling and overstimulation of metabolic pathways, resulting in the production of genotoxic products such as ROS [11].

The final stage of carcinogenesis is tumor progression. During this phase, DNA damage accumulates, and cells acquire a malignant phenotype and gain the ability to metastasize. Inflammation promotes angiogenesis, cell migration, and invasion. The tumor microenvironment plays a crucial role in facilitating these changes [12]. One of the key processes at this stage is EMT (epithelial–mesenchymal transition), which allows epithelial cells to acquire mesenchymal features, including the ability to degrade the ECM (extracellular matrix), facilitating metastasis [13].

The development of PDAC involves a series of mutations that transform from normal mucosa through precursor lesions such as PanIn (pancreatic intraepithelial neoplasia) and IPMN (intraductal papillary mucinous neoplasm) [14]. PanIn neoplasm is a lesion that occurs in small pancreatic ducts. In addition, it is the most common precursor of PDAC [14]. It is speculated that PanIn plays a role in the development of local pancreatitis, which causes epithelial damage that consequently promotes carcinogenesis [15]. Early PanIN lesions have been shown to have mutations in the *KRAS* (Kirsten rat sarcoma viral oncogene homolog) oncogene and also exhibit telomere shortening, while lesions present in PanIN with a higher degree of malignancy evidence mutations in genes such as *TP53* (Tumor Protein P53), *SMAD4* (SMAD Family Member 4), or *p16* [16,17]

Another precursor lesion of pancreatic cancer is IPMN, which can originate in the main pancreatic duct or in one of its side branches [18]. Depending on the origin of the IPMN, the incidence of associated PDAC may vary. The diagnosis of IPMN provides an opportunity to treat the immediate precursor of PDAC, although a significant proportion of patients with IPMN do not progress to invasive cancer [19].

MCNs (mucinous cystic neoplasms) are changes with a fairly low incidence [20]. They constitute about 10% of cystic pancreatic lesions [21]. MCN is a precursor lesion of PDAC with low-grade dysplasia or high-grade dysplasia [21]. The curative therapy for MCN with both low-grade and high-grade dysplasia is surgical removal of the tumor.

### 3.3. Risk Factors for Pancreatic Cancer

Environmental factors play a major role in the pathogenesis of pancreatic cancer, as only about 10% of PDAC cases are hereditary [22]. Risk factors for pancreatic cancer are classified into lifestyle modifiable and non-modifiable factors, such as age, gender, genetic factors, or ethnicity. Although the genes most frequently mutated in PDAC have been identified, the exact circumstances under which these mutations arise are not yet fully understood. Cigarette smoking and obesity are strongly associated with pancreatic cancer [23,24]. Lifestyle and environmental factors such as cigarette smoking, alcohol consumption, processed food intake, or environmental toxins are a source of molecules that induce a series of changes in DNA, leading to carcinogenesis [25]. The risk of pancreatic cancer increases with age, making it a disease predominantly of the elderly. Nearly 90% of cases are diagnosed after the age of 55, with peak incidence observed in those over 70 years of age [26]. Notably, geographic differences in pancreatic cancer incidence can be explained by differences in life expectancy by region. The Global Burden of Diseases, Injuries and Risk Factors Study 2019, which aimed to develop an assessment of key demographic indicators, showed an increase in life expectancy in recent years, where the highest was in European countries and North America, which correlate with a high incidence of pancreatic cancer [5] (Table 1, Figure 1). Another non-modifiable risk factor is ethnicity. The overall incidence of pancreatic cancer is the highest among African-Americans, followed by non-Hispanic whites and Hispanics, and the lowest among Asians [27]. The incidence of pancreatic cancer in both sexes is insignificant. It occurs with a similar frequency in both sexes; however, there is a slight predominance in men, 5.5/100,000, compared to women, 4/100,000 [17,28]. Unfortunately, the reasons for the slightly higher incidence of pancreatic cancer in men are unknown [17]. It has been suggested that the disparity can be attributed to differences in lifestyle, primarily a higher rate of smoking in men [29]. In addition, biological differences between the sexes also have an impact, as estrogen reduces the growth of pancreatic cancer [30,31,32].

Non-modifiable risk factors, including genetic factors and familial/hereditary factors, encompass mutations in the *KRAS*, *SMAD4*, *c-myc* (cellular myelocytomatosis oncogene), *TP53*, and *BRCA1/BRCA2* (Breast cancer gene 1/breast cancer gene 2) genes [33]. In addition, Lynch syndrome, Peutz–Jeghers syndrome, cystic fibrosis, and familial adenomatous polyposis represent non-modifiable factors [33].

A major modifiable risk factor for pancreatic cancer is smoking. A study conducted in Los Angeles County involving 490 patients showed that smoking at least one pack of cigarettes per day increased the risk of pancreatic cancer 5–6 times [34]. Cigarette smoke contains a diverse array of chemical compounds that cause cancer. Smoking influences the development of PDAC by inducing DNA methylation and creating DNA adducts that combine to activate mutations in the *KRAS* gene [35]. Tobacco smoke contains aldehydes, PAH (polycyclic aromatic hydrocarbons), HAA (heterocyclic aromatic amines), nitrosamines, and benzene, which bind to DNA to form DNA adducts [36]. In addition, nicotine as the main component of cigarettes has an immunosuppressive effect because it reduces the phagocyte activity of neutrophils, affecting chemotaxis [37]. As a result, smokers experience chronic inflammation, which not only contributes to increased cell proliferation, but also interferes with cell defense mechanisms, such as apoptosis or DNA repair, which protects cells from uncontrolled cell division [37].

Another modifiable risk factor is alcohol consumption. During ethanol metabolism, a series of reactions occur that increase the production of ROS, leading to oxidative DNA damage [38]. In addition, one of the products of alcohol metabolism is acetaldehyde, which, accumulating in cells, reacts with DNA, creating ICLs (interstrand crosslinks) [39]. For PDAC, cohort studies have shown a significant correlation between alcohol consumption and an increased cancer risk [40,41]. Risk factors for pancreatic cancer include pancreatitis, which can result from excessive alcohol consumption or have a hereditary basis [42].

Insulin resistance, closely associated with obesity and type 2 diabetes, is another risk factor for pancreatic cancer. It has been observed that in obese people, there is an increase in both basal and postcibal insulin concentrations in the plasma [43]. Peripheral tissues are less sensitive to insulin, so insulin uptake is reduced, and in addition, insulin uptake and removal from portal and peripheral plasma is decreased [43]. Obesity and insulin resistance may precede the onset of type 2 diabetes by many years [44]. A wide range of DNA damage has been observed in obese people, including SSBs (single-strand DNA breaks), DSBs (double-strand DNA breaks), and oxidized bases [45]. This is mainly related to chronic energy overload, which causes increased ROS production and inflammation [45]. DNA damage together with insulin resistance and chronic inflammation change gene expression, which promotes the proliferation and migration of cancer cells, resistance to apoptosis, and even tumor angiogenesis [45]. In addition, hyperinsulinemia associated with obesity causes mutations in the p53 protein, a suppressor gene increasing the proliferation and invasiveness of cancer cells [46]. Obese people often have a low intake of fruits and vegetables, which are rich in antioxidants such as carotenoids, vitamin C and E, polyphenols, or flavonoids, neutralizing ROS and protecting DNA [47]. Importantly, a huge number of carcinogenic substances enter our bodies with food. An improper diet rich in highly processed products supplies the organism with HAA and PAH, which damage DNA mostly by creating DNA adducts.

Type 2 diabetes is a consequence of an unhealthy lifestyle. Excess adipose tissue is associated with the secretion of inflammatory cytokines and chemokines by visceral adipocytes, which alters insulin signaling and may induce insulin resistance [48]. In obese people, activated macrophages also secrete pro-inflammatory cytokines, including TNF-α (Tumor Necrosis Factor) or IL-6 (Interleukin-6), which induce DNA damage at sites distant from the site of inflammation [45]. Abnormal secretion of adipokines and cytokines leads to the activation of oncogenic pathways such as NF-κB (Nuclear Factor kappa B) or PI3K/AKT (phosphatidylinositol 3-kinase/ protein kinase B) [49]. Additionally, these people have reduced secretion of adiponectin, which has anti-inflammatory and anti-angiogenic properties, alongside increased secretion of leptin, which is a mitogenic, anti-apoptotic, pro-angiogenic, and pro-inflammatory factor [49,50].

### 3.4. Diagnosis, Treatment, and Survival

Despite the significant development of medicine, there are still insufficient diagnostic methods that would enable the detection of pancreatic cancer at an early stage. It has been shown that the latency period between the onset of pancreatic carcinogenesis and the first symptoms is about 10 years [51]. However, these symptoms are non-specific such as nausea, abdominal pain, or weight loss, which are very often confused with other diseases. This contributes to a delayed diagnosis of this cancer [52]. Current diagnostic methods include ultrasonography, computed tomography, magnetic resonance imaging, positron emission tomography, and endoscopic ultrasonography [53]. These methods are sensitive enough to detect pancreatic tumors and metastases, both local and distant. Initial diagnosis using imaging methods requires histological evaluation of tissue using fine-needle biopsy [54]. Biomarkers for the early detection of pancreatic cancer, such as CA 19-9 (cancer antigen 19-9), or epigenetic markers such as ADAMTS1 (disintegrin A and metalloproteinases with thrombospondin motifs) have also become an area of intensive research. Despite the ease of determining the CA19-9 marker, it is unfortunately not specific enough for pancreatic cancer, meaning that it does not have a high predictive value [14].

The choice of treatment method depends on the stage of the disease (Figure 2).

There are four stages of pancreatic cancer. In stage I, the tumor is small and is located in the place of origin. In stage II, the cancer is locally spread or borderline resectable [17]. In stage III, the tumor is already larger and has started to spread to neighboring lymph nodes or vessels, while the last stage, IV, is indicative of distant metastases. If PanNET is diagnosed at an early stage, the prognosis for patients is optimistic [17]. In the first stage of the disease, the average 5-year survival rate after treatment is as much as 12–14% [51]. In the second stage, the 5-year survival rate lowers to 5–7%, in the third stage to 3%, and in the fourth stage to about 1% [51]. In the first and second stages, one of the methods of treatment is surgical resection, although only about 10–20% of diagnosed patients have a chance of the procedure being successful [53]. In addition to resection, radiotherapy and chemotherapy are also applied. The most commonly used chemotherapeutic agents are FOLFIRINOX (5-fluorouracil, folinic acid, irinotecan, and oxaliplatin), gemcitabine, and nab-paclitaxel [55]. The mechanism of action of 5-FU (Fluorouracil) is through the inhibition of thymidylate synthase, interfering with the synthesis of pyrimidine-thymine, which is required for the synthesis of genetic material, leading to cell death, and folinic acid helps 5-FU work more effectively and reduces its side effects [56]. Irinotecan has cytotoxic effects by inhibiting DNA topoisomerase I and, preventing DNA strand ligation leading to DNA double-strand breaks and cell death [56]. Oxaliplatin, on the other hand, is converted into active derivatives that bind to the guanine and cytosine groups of DNA, resulting in DNA cross-linking that inhibits DNA synthesis and transcription [56]. Paclitaxel’s mechanism of action is to induce mitotic arrest, leading to cell death, while gemcitabine’s mechanism of action is to penetrate the DNA double helix, resulting in the disruption of synthesis and leading to cell death [57]. After resection, patients are often given adjuvant systemic chemotherapy, but it has negative, toxic effects on patients. The median 5-year survival rate after such surgery and chemotherapy is approximately 20%, although the tumor recurrence rate remains at 70% [58,59,60]. The third stage of the disease strikes about 30% of patients. Unfortunately, more than 50% of all patients are diagnosed at the last stage of the disease [61]. In stage III patients, in some cases surgery is preceded by neoadjuvant chemotherapy, which shrinks the tumor, and then the surgery is carried out, while stage IV patients receive chemotherapy alone. This is because the pancreas is located close to the vascular system, and patients in this stage have distant metastases, making resection either impossible or not beneficial for the patient [62]. Unfortunately, both systemic and neoadjuvant chemotherapy are toxic, so alternative ways of delivering drugs to the patient in the least harmful way are sought. An alternative to traditional therapeutic drugs is the use of nanoparticles carrying drugs targeted at the tumor tissue, thus reducing the side effects that often accompany systemic chemotherapy [63]. Examples of such nanoparticles include polylactic and glycolic acid, human and bovine albumin nanoparticles, and nanoparticles prepared from natural polymers [63]. These particles can transfer chemotherapeutic agents or drugs, an example of which is albumin-bound paclitaxel, available under the trade name Abraxane^®^ [64]. Interestingly, in mouse xenograft models of pancreatic cancer, this drug reduced pancreatic cancer stroma [64]. In addition to greater treatment efficiency, such nanoparticles are non-toxic, non-immunogenic and readily biodegradable. Taken together, the diagnostic options for pancreatic cancer remain limited and rely primarily on imaging techniques, such as computed tomography and magnetic resonance imaging. To date, no specific biomarkers have been identified that would allow for early-stage detection of the disease. Therapeutic strategies are likewise constrained, with treatment choices largely determined by the diagnosis stage. Among the available approaches, surgical resection combined with chemotherapy is recommended in early stages, while neoadjuvant chemotherapy and, where feasible, partial resection may be considered.

Unfortunately, pancreatic cancer patients are usually placed under palliative care because other available treatments are impossible or ineffective. In addition, pancreatic cancer is aggressive and chemoresistant, which makes it even more difficult to treat.

## 4. DNA Damage

Living organisms are constantly exposed to a number of harmful factors that cause DNA damage. DNA damage can be defined as a modification of DNA that interferes with its proper functioning during transcription or replication [65]. Every human cell experiences about tens of thousands of instances of DNA damage per day, and most of them are the result of physiological cellular processes [66]. For this reason, the body has developed a number of mechanisms to protect against DNA damage, including DNA repair and the antioxidant system. DNA damage causes cell cycle arrest or cell death [67].

We can subdivide DNA damage into endogenous, induced by cellular processes, and exogenous, which is caused by environmental factors. A classification by type of damage can also be used. This subdivision includes SSBs and DSBs, nitrogen base modifications, DNA adducts, and pyrimidine dimers [68,69]. The classification of DNA damage is shown in Table 2.

### 4.1. Alkylation of Bases

DNA alkylation is a process in which an alkyl group is added to specific DNA bases. Examples of alkyl groups include the following: methyl group, butyl group, isopropyl group, or ethyl group. The sources of alkylating agents can be both endogenous, arising as byproducts of oxidative damage or from cellular methylation donors, and also of exogenous origin, such as tobacco smoke [70]. Transferring an alkyl group to biological molecules, including DNA, changes their structure and may disrupt their functions [70]. The most common alkylation products include O^2^-alkylthymine, O^4^-alkylthymine, O^6^-methylguanine, and O^6^-ethylguanine [71]. For instance, O^6^-methylguanine is a procarcinogenic DNA adduct that has been linked to the formation of GC < AT mutations in the *KRAS* oncogene [72].

Most DNA alkylation products are mutagenic and cytotoxic and an example is N^3^-methyladenine [70]. Other instances include N^1^-methyladenine and N^3^-methylcytosine, which are unable to base pair and subsequently block DNA replication [73]. It is speculated that the most mutagenic alkylation products are O^6^-alkylguanines and O^4^-alkylthymines. During replication, O^6^-alkylguanines misincorporate thymine instead of cytosines, leading to G > A transition mutations, while O^4^-alkylthymines misincorporate guanines instead of adenines, leading to T > C mutations [74]. Additionally, O^6^-alkylguanines, due to their cytotoxicity, cause mispairing after replication, resulting in ineffective DNA mismatch repair (MMR), which leads to the formation of DSB and ultimately cell death [75].

### 4.2. Purine and Pyrimidine Dimers

UVR (ultraviolet radiation) is one of the factors that induce mutagenic and cytotoxic DNA damage. A common source of UV radiation is sunlight [76]. UVR can be divided into three groups, differing in wavelength: UV-A, UV-B, and UV-C. UV-B radiation is the most dangerous to humans. Photoproducts resulting from UVR include 6-4PP (6-4 pyrimidine-pyrimidone photoproducts), CPD (cyclobutene pyrimidine dimer,) or Dewar valence isomers [77]. Dewar valence isomers are the least common photoproducts and are formed when an additional photon is absorbed by 6-4PP [78]. CPDs are formed by cycloaddition between two pyrimidine bases [79]. These DNA lesions distort the helix, posing an obstacle to RNA and DNA polymerases, and error-prone bypass of CPD lesions during replication [80]. In turn, the formation of 6-4PP occurs during a two-step process, where two intermediate products can be formed [81]. CPDs constitute about 75% and 6-4PP 25% of the total UV-mediated DNA damage products [82].

Another factor that leads to DNA damage is ionizing radiation (IR). IR is high-energy radiation that releases electrons from atoms, generating ions which can break covalent bonds [83]. IR includes gamma rays, X-rays, alpha and beta particles, and neutrons [83]. This type of radiation causes mainly DSBs. In addition, radiation causes oxidation of proteins and lipids by ROS, resulting in the formation of AP (apurinic/apyrimidinic) sites [84,85]. IR reaches healthy tissue, where it can contribute to the formation of chromosomal aberrations, increasing the risk of cancer [86]. Due to its ability to damage DNA, IR is used in radiotherapy to reduce tumor mass or eliminate residual tumor cells by exposing the tumor to IR [83].

IR radiation does not significantly increase the development of pancreatic cancer. UV radiation is primarily a major factor in the development of skin cancer. DNA damage caused by radiation does not affect the development of pancreatic cancer. Interestingly, studies have been conducted to examine the relationship between UV radiation, vitamin D, and pancreatic cancer. These studies have shown that fair-skinned people with a more sun-sensitive phenotype have a lower risk of pancreatic cancer compared to dark-skinned people [87].

### 4.3. DNA Adducts

DNA adducts are fragments of DNA that are covalently bonded with a chemical [88]. If they are not removed by the DNA repair system, they cause miscoding during replication. Sources of substances that form adducts include cigarette smoke, alcohol metabolism or food, and the way it is processed (fried, grilled) [89]. Examples include formaldehyde, acetaldehyde, NNK (nicotine-derived nitrosamine ketone), and NNN (N-nitrosonornicotine) [90]. In addition, safrole and estragole are found in herbs or spices, which are sources of N-nitrosamines [90]. Another example is aflatoxin, which is a mycotoxin produced by fungi of the *Aspergillus* genus. Aflatoxins enter the body through the consumption of contaminated products, such as grains [91].

Li et al. conducted a study to investigate the role of carcinogen exposure in PDAC [92]. They measured total aromatic and lipid peroxidation-related DNA adducts in pancreatic tumor tissues and adjacent noncancerous pancreatic tissues obtained from patients undergoing surgical resection [92]. The study demonstrated that tumor tissues contained significantly higher levels of DNA adducts than adjacent normal tissues. Although specific chemical identities of the adducts were not determined, these DNA modifications likely derived from PAH from cigarette smoke and HCA (heterocyclic amines) from diet, as well as from lipid peroxidation products such as MDA (malondialdehyde) [92]. Additionally, Li et al. correlated DNA adduct levels with *KRAS* mutations and genetic polymorphisms in DNA repair genes, suggesting a link between environmental exposure, DNA damage, and pancreatic carcinogenesis [92]. Similarly, Wang et al. investigated the potential impact of environmental and dietary carcinogens on pancreatic carcinogenesis [93]. Specifically, they examined DNA adducts formed by PAHs derived from cigarette smoke and lipid peroxidation products, mainly MDA, which can result from oxidative degradation of polyunsaturated fatty acids [93]. Using the ^32P-postlabeling method, the authors quantified aromatic and lipid peroxidation-related DNA adducts in pancreatic tumor tissues and adjacent noncancerous tissues obtained from patients with pancreatic cancer [93]. They observed significantly higher levels of MDA-DNA adducts in cancerous tissues compared with adjacent normal tissues [93]. These findings suggest that DNA damage resulting from both PAH-derived and lipid peroxidation-derived adduct formation may contribute to pancreatic carcinogenesis [93].

### 4.4. ICL

ICLs are linkers between DNA strands that prevent their separation [90]. ICLs can arise both through chemical bonding and as byproducts of cellular metabolism [94]. These lesions prevent transcription and replication by inhibiting the separation of DNA strands, which is accomplished by coordinated chemical reactions on opposing strands [95]. The covalent bonds that form either between a base on opposite strands or between a DNA base and a chemical compound are irreversible. Additionally, cross-links can form between bases within the same DNA strands [95].

DPCs (DNA–protein cross-links) are a distinct type of DNA adducts in which proteins become covalently attached to DNA following exposure to physical or chemical cross-linking agents [96]. DPCs can be induced by exogenous factors like IR, UVR, and chemotherapeutics, as well as endogenous sources like AP sites, aldehydes, and topoisomerases [97,98]. These cross-links significantly disrupt DNA replication, transcription, recombination, and repair [96]. Notably, this type of DNA damage is associated with factors that elevate the risk of pancreatic cancer. For example, acetaldehyde, which is a byproduct of alcohol metabolism, can induce formation, contributing to genomic instability and carcinogenesis.

### 4.5. Oxidative DNA Damage

One of the primary causes of DNA damage is oxidative stress. This is a condition in which the balance between the amount of ROS and the body’s ability to remove them through antioxidants is disturbed [99]. The sources of oxidative stress can be both endogenous and exogenous [100]. ROS are generated as a result of altered oxygen pressure, exposure to chemical compounds, radiation (e.g., UV, IR), metabolic processes, or inflammation [101]. ROS are molecules containing one unpaired electron, which makes them highly reactive [102]. Most ROS are derived from oxygen and nitrogen. Examples include the superoxide anion radical (O_2_•^−^), the hydroxyl radical (•OH)—one of the most reactive and toxic ROS—ozone (O_3_), and hydrogen peroxide (H_2_O_2_), which is formed from the superoxide anion in an enzymatic reaction catalyzed by superoxide dismutase.

Oxidative DNA damage generated by ROS leads to base modification that causes mutations and contributes to the development of numerous diseases, including cancer [103]. One of the DNA bases most frequently subject to oxidative damage is guanine. Guanine has a low oxidation potential, making it very susceptible to singlet oxygen, resulting in the formation of 8-oxoG (8-oxo-7,8-didydroguanine) [104,105]. Such a modified purine is inserted during replication and incorrectly pairs with adenine, which results in the disruption of cell functions by changing the way proteins or transcription factors bind to DNA [106,107]. Moreover, 8-oxoG changes the secondary structure of DNA, affecting replication and gene regulation [108]. It is considered one of the most widely used biomarkers of ROS-induced DNA damage [109].

Oxidative DNA damage is one of the major contributors to the development of pancreatic cancer [110]. Mohamadkhani et al. demonstrated that patients with pancreatic cancer had significantly higher levels of 8-OHdG (8-hydroxy-2′-deoxyguanosine) in peripheral blood leukocytes, a marker of oxidative DNA damage, compared to healthy individuals [110]. Tobacco smoke—a leading risk factor for pancreatic cancer—causes overproduction of ROS, thereby promoting oxidative stress. In addition, smoking promotes chronic inflammation, which further enhances ROS generation. Another key factor is hyperglycemia, a hallmark of diabetes, which promotes ROS overproduction and oxidative DNA damage, contributing to carcinogenesis.

### 4.6. DNA Strand Breaks

Other types of DNA damage include SSBs and DSBs [111]. SSBs are a part of endogenous DNA metabolic processes [111]. The main source of SSBs is oxidized deoxyribose, which after abstraction of a hydrogen atom can break down into SSBs containing a fragmented deoxyribose 3′ end [111]. Additionally, SSBs are generated during BER (Base Excision Repair) through the activity of specific DNA glycosylases such as OGG1 (8-oxoguanine DNA Glycosylase) or NEIL-3 (Nei Like DNA Glycosylase 3) or through the activity of APE1 (human apurinic/apyrimidinic endonuclease) [112]. Another physiological source of SSBw is the activity of DNA topoisomerase, which creates DNA breaks during replication [111].

DSBs are induced by environmental factors such as ionizing radiation, anticancer drugs (cisplatin), or radiomimetic compounds (phleomycin) [113]. They also result from independent breaks in the sugar-phosphate backbone on opposing strands of the DNA molecule, typically occurring 10–20 base pairs apart [114]. Moreover, DSBs are caused by impaired DNA replication, transcription, and recombination [115]. These breaks may occur during attempts to repair oxidized DNA bases when they occur simultaneously on opposing strands [116]. DSBs destabilize genomic DNA by causing inversion, deletions, and chromosomal translocations [117]. Such alterations can promote the proliferation of cancer cells, as the resulting mutations may confer clonal advantages, contributing to the uncontrolled regulation of cell growth and progression to neoplastic phenotypes [114].

Li YH et al. conducted in vitro studies on 18 PDAC cell lines to evaluate the role of DNA repair in PDAC. They observed that PDAC cells are characterized by elevated levels of γH2AX foci, which is a marker of DNA damage, primarily DSBs [118]. They also showed that inhibition of the NHEJ repair pathway by inhibiting DNA-Pkcs, a key subunit of the NHEJ pathway, further increases DSB accumulation [118]. Next, the response of PDAC cells to DNA damage after IR was examined at different time points by measuring γH2AX foci [118]. It was found that DNA repair was significantly impaired by DNA-Pk inhibition [118]. Importantly, NHEJ inhibition sensitizes PDAC cells to IR, which was verified by DNA-Pkcs inhibition followed by irradiation at doses of 2, 4, and 6 Gy (grey) [118]. These findings highlight a potential therapeutic window for exploring synthetic lethality. If PDAC cells already exhibit HR defects, additional inhibition of NHEJ may result in the accumulation of unrepaired DSBs, ultimately triggering cancer cell death. Therefore, comprehensive profiling of DDR in tumors may inform personalized treatment strategies targeting synthetic lethality.

## 5. A General Overview of the DDR Pathways

Maintaining genome stability is crucial for all cells, as any damage can disrupt biological processes [119]. For this reason, cells have evolved a variety of repair mechanisms to prevent the accumulation of DNA damage. The DNA damage response is a complex signaling network based on several repair pathways, including BER, NER (nucleotide excision repair), NHEJ (non-homologous end joining), HR (homologous recombination), and MMR [120]. Proteins involved in DDR can be classified into three groups: sensors, mediators, and effectors. Sensor proteins detect DNA abnormalities and initiate the recruitment of other factors involved in repair. Examples of sensors include DNA glycosylases, the MSH2/MSH6 (MutS Homolog 2/MutS Homolog6) complex, XPC (Xeroderma pigmentosum group C), Ku70/80, or MRNs. Mediator proteins participate in repairing damage and include APE1, PCNA (Proliferating cell nuclear antigen), ERCC1 (DNA Excision Repair Protein1), TFIIH (Transcriptional factor IIH), DNA PKcs (DNA-dependent protein kinase catalytic subunit), or BRCA1/BRCA2. Effector proteins consist primarily of various polymerases and ligases responsible for synthesizing new DNA strands or joining DNA strands. Each repair pathway consists of three stages, where the first stage is the recognition of damage, next is the excision or processing of the damaged chain, and the final stage is repair. DDR is mainly mediated by proteins from the PI3K family, such as ATM (ataxia-telangiectasia mutated), ATR (Ataxia Telangiectasia And Rad3-Related Protein), and DNA-PK, as well as members of the PARP (Poly (ADP-ribose) polymerase) family [121]. These sensor proteins are the first to recognize specific types of DNA damage. Upon detection, ATM and ATR proteins phosphorylate mediator proteins to amplify the DNA damage response [122]. The targets of ATM and ATR are most often the protein kinases CHK1 (checkpoint kinase1) and CHK2 (checkpoint kinase2) [123]. This results in the formation of the ATM-CHK2 and ATR-CHK1 pathways. As a result, the entire DDR signaling cascade is activated, involving various cell cycle checkpoints, repair systems, apoptosis regulators, and other molecules such as ligases, polymerases, and helicases, all coordinating to repair damaged DNA and maintain genome integrity [124]. A summary of the DNA repair pathways is provided in Figure 3.

### 5.1. BER

DNA damage such as base modifications (alkylation, oxidation, and deamination) is repaired by the BER pathway. This repair mechanism operates mainly in the nucleus, but also functions to a lesser extent in mitochondria [125].

BER is initiated by one of several specific DNA glycosylases, such as NEIL1 (Endonuclease 8-like 1), UNG (Uracil-N-glycosylase), or SMUG1 (Single-Strand-Selective Monofunctional Uracil-DNA), as shown in Figure 4 [125]. These glycosylases recognize the damaged base and hydrolytically cleave the N-glycosol bond that links the base to the sugar-phosphate backbone, thereby generating an AP site [126]. The AP site is then cleaved on its 5′side by the APE1 endonuclease, which hydrolyzes the DNA backbone, creating 3′hydroxyl site and filling the gap [127,128]. In BER, the main polymerases involved are DNA polymerase β, δ, ε, and λ. The final step involves sealing the DNA ends with the help of DNA ligases, primarily DNA ligase IIIα together with the XRCC1 (X-ray Repair Cross Complementing 1) [129]. Depending on whether a short fragment (at most 2 nucleotides) or a fragment of up to 13 nucleotides is being repaired, different molecules are involved [130]. In the SN-BER pathway (short-patch BER), the key polymerase is DNA polymerase β, which incorporates a single nucleotide, and then DNA ligase III seals the incision. In the LP-BER pathway (long-patch BER), DNA polymerases δ and ε in combination with PCNA and RFC (Replication factor C) factors incorporate multiple nucleotides, and the resulting incision is sealed by DNA ligase I [127].

### 5.2. NER

NER is one of the key DNA repair pathways responsible for removing various types of base damage, shown in Figure 5, [131]. This pathway most often responds to the mutagenic effects of environmental factors, such as UVR, IR, and base alkylation [132]. NER is capable of removing bulky DNA lesions, but also removes intrachain cross-links [133].

The NER pathway has two subpathways: GG-NER (global genomic NER), which is initiated by the recognition of DNA helix distortions induced by damage, and TC-NER (transcription-coupled NER), which is activated by the retention of RNA polymerase II at the site of damage [134]. The XPC-RAD23-CENT2 (Xeroderma pigmentosum group C) complex plays a central role in GG-NER by recognizing DNA damage [135]. Once this complex binds to the damage site, it recruits the TFIIH complex, which contains two helicases: XPD (Xeroderma pigmentosum group D) and XPB. In vitro studies have shown that TFIIH scans DNA in the 5′-3′ direction for damage blocking of the XPD helicase, which is an essential factor required for damage verification [136]. Following damage verification, specific endonucleases XPF-ERCC1 and XPG excise the damaged DNA segment [137]. This results in the removal of a single-stranded break of about 20 nucleotides in length, which causes XPC to mark the incision site and then XPA (Xeroderma pigmentosum group A), XPG (Xeroderma pigmentosum group D), and RPA (Replication Protein A) bind to the tension site [134]. XPA is considered a key regulator of the NER pathway, as it enhances TFIIH damage verification and interacts with several NER components to ensure accurate repair [138]. The gap formed after excision initiates DNA synthesis, which is coordinated by RPA and XPG [139]. The final synthesis is carried out by the PCNA protein, RFC and DNA polymerase δ, and DNA polymerase ε, and at the end, DNA ligase I fills the resulting gaps [134]. The activity of NER proteins, as well as DSB repair proteins, is regulated by post-translational modifications including ubiquitylation, sumoylation, phosphorylation, acetylation, and poly (ADP-ribosyl)ation [134].

The TC-NER pathway is specialized for the selective repair of transcription-blocking lesions, thereby allowing transcription to resume efficiently [134]. TC-NER is triggered when RNA polymerase II stalls at a DNA lesion, leading to the recruitment of CSA (Cockayne syndrome A) and CSB (Cockayne syndrome B) proteins. Initially, CSB binds to the stalled RNA polymerase II, facilitating the recruitment of CSA. In turn, CSA acts on the E3 ubiquitin ligase complex, which facilitates the ubiquitination of CSB and RNA polymerase II [140]. At the same time, TFIIH is recruited to the damage site. Despite some differences, many elements of TC-NER overlap with GG-NER. After DNA damage is recognized, the TFIIH factor is recruited, which together with RPA verifies the damage [141]. Then, repair endonucleases such as ERCC1-XPF (Xeroderma pigmentosum group F) or XPG cut the damaged strand, and then DNA polymerases fill in the gap, until finally DNA ligase I or the ligase IIIα-XRCC1 complex seals the DNA backbone [141].

### 5.3. NHEJ

The NHEJ repair pathway is responsible for repairing the majority of DSBs. Importantly, this pathway does not require a homologous template for repair, making it active throughout all phases of the cell cycle [142]. However, NHEJ is less precise than HR and therefore error-prone.

At the moment of DNA breakage, the DNA ends are bound by the Ku70/80 heterodimer, a basket-shaped molecule whose arms form a ring that tightly encircles the DNA, enabling it to bind directly to the DNA ends, as shown in Figure 6 [143,144]. The formation of the Ku70/80 basket is a critical step in the NHEJ process. Additionally, the Ku heterodimer has been shown to bind to the sugar-phosphate backbone of DNA, emphasizing its ability to attach to DNA in a sequence-independent manner [145]. Ku directly recruits major NHEJ factors such as DNA-PKcs, XRCC4 (X-ray repair cross complementing 4) protein or APLF (Aprataxin And PNKP Like Factor) [145]. Ku70/80 binding enables the PI3K-associated DNA-PKcs kinase to recognize the DNA-Ku complex and form an active DNA-PK complex [146]. The binding of DNA-PKcs to this complex causes the Ku heterodimer to translocate along the dsDNA strand, thereby activating DNA-PKcs kinase [145]. Subsequently, Artemis—a specific nuclease—binds to DNA-PK, activating the complex and allows it to trim the DNA ends [147]. In later steps, Ku interacts with a complex consisting of DNA ligase IV and XRCC4, facilitating coordination among these enzymes to prepare the DNA ends for final ligation. To complete the repair process, polymerase µ is required, as it brings the two DSB ends together and contributes to end processing and fill-in synthesis [148].

### 5.4. HR

Homologous recombination is a key pathway responsible for the repair of DSBs. In addition to its role in DSB repairing, HR is crucial for maintaining genome stability and ensuring accurate genome duplication [149]. This pathway is active during the S and G2 phases of the cell cycle, i.e., after DNA replication [150].

As with all DNA repair mechanisms, the first step in HR involves the recognition of DNA damage by specific proteins. ATM kinase is a protein that recognizes and recruits to the damage site. A central component of the HR machinery is the MRN complex, composed of MRE11, RAD51, and NBS1, as shown in Figure 7 [151,152]. This complex interacts with CtIP protein (c-terminal binding protein), initiating 5′-3′ resection of DNA ends, which generates ssDNA (single-strand DNA) overhangs at both break sites [153]. This resection is further enhanced by nucleases such as EXO1 (Exonuclease 1) and DNA helicase 2 which help unwind the DNA. Once resection occurs, the ssDNA is rapidly coated by RPA (Replication Protein A), which stabilizes the ssDNA region and prevents secondary structure formation [153]. Next, BRCA1 recruits BRCA2, which serves as a mediator protein. BRCA1 facilitates the replacement of RPA with RAD51, a recombinase that binds to ssDNA and forms a nucleoprotein filament essential for homology search and strand invasion [153]. RAD51 then mediates the invasion of the sister chromatid, forming a structure called a D-loop, which causes DNA synthesis by polymerase δ, starting from the 3′end of the invading strand [154,155]. The final step is the joining of the chains by DNA ligase 1.

### 5.5. MMR

MMR is a highly conserved post-replication repair pathway that corrects mismatched nucleotides or small insertion–deletion loops generated by DNA polymerase during replication [156]. DNA replication is a faithful process, with spontaneous mutations occurring at a frequency of 1 in 10^9^–1 in 10^10^ base pairs [157]. The selection of correct nucleotides during base incorporation and the proofreading activity of DNA polymerases result in an error rate of about 10^−7^ base pairs per genome [157]. The MMR pathway acts on replication repair errors that escape correction by DNA polymerases, thereby significantly increasing the overall fidelity of DNA replication, as shown in Figure 8.

The MMR process occurs in three main stages. First, damage recognition is performed by protein complexes such as MutSα and MutSβ which detect mismatches and identify the insertion–deletion loop site [158]. MutSα is a heterodimer composed of MSH2 and MSH6, and it binds to both base mismatches and single-base insertion–deletion loops [159]. In contrast, MutSβ is a heterodimer consisting of MSH2-MSH3 (MutS Homolog 3), with lower affinity for mismatched bases and higher specific affinity for insertion–deletion loops involving several nucleotides [159]. These complexes migrate to the mismatch site and bind to DNA, forming a so-called sliding clamp structure [119].

In the second stage, binding to MSH2 and MSH6 induces conformational changes in the MutS complex, which facilitates the recruitment of the MutL homolog, a complex formed by MLH1 and PMS2 [156]. This event initiates the recruitment of several proteins involved in the repair process. One of these is EXO1, an exonuclease that excises the newly synthesized DNA strand containing the error. This creates a gap that can subsequently be filled by DNA polymerases, forming the final stage of repair. This repair is also mediated by PCNA, which interacts with both MLH1 and MSH2, and plays a key role in DNA initiation and resynthesis during repair [156]. Another essential factor in MMR is RPA (replication protein A), an ssDNA binding protein. RPA binds to the nicked heteroduplex DNA, enhances mismatch-induced excision, and protects the exposed ssDNA during processing, thereby facilitating DNA resynthesis [156].

It is important to emphasize that alterations in these pathways, especially mutations in DDR genes, have significant implications for PDAC development and progression. Understanding the landscape of DDR gene mutations provides insight into the molecular mechanisms driving genomic instability in PDAC, as well as potential therapeutic strategies. Both germline and somatic mutations have been identified in key DDR genes such as *BRCA1*/*BRCA2*, *PALB2*, *RAD51*, *FANCB*, *ARID1A*, and *ATM* [160,161,162]. Dysfunction of these genes and their encoded proteins leads to defects in repairing DSBs, particularly through HR deficiency [163]. In PDAC, such defects contribute to genomic instability, facilitating the accumulation of mutations and tumor progression [164]. For example, mutations in *BRCA2* and *PALB2* impair the ability to accurately repair DSBs, affecting tumor cell proliferation and survival [164]. Additionally, mutations in *ATM* and *RAD51* disrupt DNA damage signaling and checkpoint control, which may lead to compromised cell cycle regulation and defective apoptosis [165].

## 6. DDR and PDAC

It is well known that PDAC is a very aggressive cancer. Its incidence is steadily increasing and it is speculated that by 2030, the number of pancreatic cancer cases will surpass those of colon cancer [166]. The development of pancreatic cancer has been linked to both lifestyle factors (such as obesity, diabetes, and smoking) and genetic alterations (including mutations in *KRAS* or *TP53* genes). However, environmental factors appear to have a greater impact on cancer development [167]. Despite advances in science, pancreatic cancer remains a significant challenge in both research and clinical practice. The mechanisms underlying its rapid progression and invasiveness are still not fully understood.

To investigate the mechanisms behind the aggressive nature of PDAC, scientists have performed global meta-analyses of microarrays to determine which signaling pathways are disrupted. For this purpose, Jones et al. compared over 23,000 transcripts representing 20,000 genes from 24 samples taken from patients with PDAC [168]. The results of this analysis identified disruptions in 12 major signaling pathways, including apoptosis, Hedgehog signaling, KRAS signaling, and DNA damage control, with involvement of genes such as *TP53*, *EP300*, and *RANBP2* (RAN Binding Protein2) [168]. These findings indicate that DNA repair defects are among the key pathways involved in pancreatic cancer development and progression.

Waddell et al. performed whole-genome sequencing and copy number variation analysis of 100 samples from PDAC patients [163]. Using MutSig analysis, they demonstrated that chromosomal rearrangement results in mutation of the *KRAS*, *TP53*, and *SMAD4* genes [163]. Chromosomal rearrangements are a common group of mutations that lead to genomic instability, thereby promoting carcinogenesis [169]. Based on the frequency and pattern of structural rearrangements, PDAC subgroups were defined as follows: subtype 1 (stable), subtype 2 (locally rearranged), subtype 3 (diffuse), and subtype 4 (unstable) [163]. These subgroups have clinical utility and therapeutic implications. Notably, the unstable subtype accounts for approximately 14% of PDAC cases. Tumors of this subtype exhibit a high number of structural variations associated with DNA repair defects, further supporting the link between DNA repair and PDAC.

Some mutations, such as *ATM* loss, appear even in precancerous lesions like PanIN or IPMN [170,171]. Russell et al. showed in a mouse model of PDAC that *ATM* loss significantly increased the number of proliferative precursor lesions and enhanced EMT (epithelial–mesenchymal transition), with concomitant shorter mouse survival [171]. These findings demonstrate that *ATM*, one of the key DDR factors, has a significant role in the maintenance of genome integrity.

Buchber et al. conducted mutation profiling of samples taken from PDAC patients. For this purpose, they collected material from patients and the profiling results revealed mutations not only in key genes such as *KRAS*, *TP53*, and *SMAD4* but also in genes involved in homologous repair: *ARID1A* (AT-rich interactive domain-containing protein 1A), *ATR*, *ATM*, *RAD51B*, *BRCA2*, *PALB2* (Partner and localizer of *BRCA2*) and *CHEK2* (Checkpoint kinase 2), *NBN* (Nibrin), *RAD50*, *RAD51*, *FANCA* (FA complementation group A), *FANCD2* (FA complementation Group D2), and *FANCI* (FA complementation Group I) [172]. *BRCA1* and *BRCA2* genes are known to play a crucial role in the HR pathway, which is responsible for repairing DSBs. Cells deficient in *BRCA1*/*BRCA2* exhibit insufficient HR activity—a phenomenon referred to as “BRCAness”. This condition results in the accumulation of DSBs, leading to genomic instability, and HR repair deficiency may also confer sensitivity to some DNA damaging agents, resulting in chemoresistance to therapeutics used in the treatment of PDAC, such as platinum-based chemotherapeutics and PARP inhibitors [173,174]. A study conducted by Tadehara et al. on 92 patients with PDAC showed that 6 of them (6.5%) were BRCAness-positive. However, no significant differences in overall survival or progression-free survival were observed between BRCAness-positive and BRCAness-negative groups [175]. The same research team demonstrated that the KP-2 cell line, a BRCAness-positive pancreatic cancer tumor line, was more sensitive to cisplatin and olaparib compared to BRCAness-negative cell lines [175]. This suggests that BRCA status may serve as a potential biomarker for treatment selection in PDAC.

Researchers also analyzed genetic mutations across three cohort studies of familial pancreatic cancer, encompassing a total of 735 samples. The most common mutations were found in the *BRCA1*, *BRCA2*, *ATM*, *PALB2*, and *CDKN2A* genes, further highlighting the involvement of DDR-related genes in pancreatic cancer [176,177,178]. Moreover, several hereditary cancer predisposition syndromes are associated with monoallelic-dominant inherited autosomal mutations that predispose to PDAC [170]. Among them are Peutz–Jeghers syndrome with a mutation in the *STK11* gene and Lynch syndrome caused by mutations in the *MSH1* and *MSH2* genes, and an additionally increased risk of developing PDAC occurs in familial adenomatous polyposis (mutations in the *APC* gene), familial atypical multifocal melanoma syndrome (mutations in the *CDKN2A* gene), Li–Fraumeni syndrome (with mutations in the *TP53* gene), or hereditary pancreatitis (*PRSS1* and *SPINK1* mutations) [170]. Interestingly, 4–14% with apparently sporadic pancreatic cancer have also been found to carry germline mutations, despite a negative family history [179]. In this group, the most frequently mutated gene was *ATM*, whose mutations predispose to an increased risk of many cancers, and in the case of PDAC, this frequency is even three times higher. This list includes genes such as *BRCA1*, *BRCA2*, or *PALB2*, encoding a protein interacting with the BRCA2 protein which is necessary for DSB repair [160,162,179,180,181]. In addition, the list also includes *MLH1*, *MLH2*, and the TP53 protein [182].

The role of DDR in PDAC is being studied not only in in vitro models, but also in animals. Rowley et al. developed a mouse model with PDAC-specific *KRAS* and *TP53* mutations with *BRCA2* inactivation [183]. Histological analysis of mouse pancreatic tissues revealed that *BRCA2* inactivation promoted the development of precancerous lesions and pancreatic tumors [183]. In addition, cancer cells were isolated from mice and then tested for sensitivity to chemotherapeutic agents, demonstrating that BRCA2-deficient pancreatic tumors exhibit increased sensitivity to cisplatin and PARP inhibitors [183]. Similarly, Drosos et al. investigated how *ATM* gene deletion affects a mouse model of pancreatic cancer with oncogenic *KRAS* expression [184]. Their research showed that partial or complete *ATM* deficiency interacts with the *KRAS* gene to promote highly metastatic pancreatic cancer and also leads to permanent DNA damage in precancerous lesions and primary tumors [184]. Other studies investigating *ATM* deficiency in a mouse model of PDAC were conducted by Perkhofer et al. [185]. In this study, ATM-deficient mice were subcutaneously implanted with PDAC cells to induce tumor formation. Once tumors had developed, the mice were treated with gemcitabine, olaparib, or a combination of both drugs [185]. Analysis of the isolated tumors revealed that a loss of *ATM* accelerated PDAC progression and was associated with enhanced cancer cell stemness and EMT [185]. Furthermore, *ATM* deletion impaired DDR, as evidenced by a sixfold increase in γH2AX and 53BP1 foci compared to control tumors [185]. Since *ATM* deficiency compromises HR repair, the authors further explored the effect of PARP inhibition in this context. PDAC cells isolated from ATM-deficient tumors were treated in vitro with olaparib and gemcitabine in a clonogenic survival assay [185]. This treatment resulted in significantly reduced colony formation compared to controls [185].

DDR signaling in PDAC is currently under investigation in clinical trials. DDR-deficient PDAC subtypes, particularly those harboring *BRCA1/BRCA2* mutations, have shown sensitivity to platinum-based chemotherapy and PARP inhibitors such as olaparib [186]. In a non-randomized phase II clinical trial, patients with advanced PDAC carrying mutations in at least one of the following DDR genes, *ATM*, *RAD51*, *ARID1A*, *PALB2*, *FANCB*, or *BRCA1/BRCA2*, were treated with olaparib [186]. The study demonstrated that these patients had significantly longer median progression-free survival compared to historical controls, especially those who had shown prior sensitivity to platinum agents [186]. In another clinical study, the therapeutic relevance of DDR deficiency was further explored through retrospective analysis of metastatic PDAC patients treated with FOLFIRINOX [187]. Sehdev et al. observed that patients with germline or somatic mutations in DDR genes exhibited significantly improved overall survival compared to non-carriers [187]. These findings support the prognostic and potentially predictive value of DDR mutations in selecting treatment regimens involving DNA damaging agents [187].

These findings highlight the ongoing need to identify additional DDR-related targets and combination strategies to enhance treatment outcomes. Targeting multiple DDR-related pathways could help overcome compensatory repair mechanisms and increase tumor vulnerability to genotoxic agents, offering a future direction to improve survival in this highly aggressive cancer type. Two complementary strategies have emerged for exploiting synthetic lethality in PDAC, targeting HR-deficient tumors with PARP inhibitors and sensitizing HR-proficient tumors by inhibiting NHEJ. Comprehensive analysis of DDR related mutations and functional defects is thus essential to guide personalized therapy, maximize DNA damage accumulation, and promote selective tumor cell death. A summary of clinical trials related to DDR in PDAC is provided in Table 3.

## 7. Discussion

Pioneering research on DDR began in the 1960s with the discovery of direct reversal repair—specifically, the photoreactivation of separate cyclobutane pyrimidine dimers induced by UVR exposure [194,195]. Over the years, further breakthroughs have been made, such as the identification of uracil DNA glycosylase, which was the first identified DNA repair enzyme, and the discovery of DDR-related proteins including ATM, CHK1, and DNA-PKc. The first reports linking DDR to carcinogenesis date back to 1969, when Jim Cleaver connected a predisposition to skin cancer xerodorma pigmentosum to unpaired DNA damage, which was attributed to mutations in NER genes [194]. In 2015, the Nobel Prize in Chemistry was awarded to Paul Modrich, Tomas Lindahl, and Aziz Sancar for their groundbreaking studies on DNA repair.

Although more than 60 years have passed since DDR was first linked with carcinogenesis, our understanding of its role in pancreatic cancer remains limited.

Although recent progress has been made in the treatment of PDAC, mortality rates remain high due to the limited effectiveness of current therapies and the small proportion of patients who qualify for them. One of the biggest challenges is to improve overall survival and enable early-stage diagnosis (stages I and II). Environmental factors associated with the Western lifestyle—such as a physical inactivity, smoking and obesity, often accompanied by insulin resistance and type 2 diabetes—appear to have a dominant influence on PDAC development. Hyperglycemia, a hallmark of diabetes, leads to oxidative DNA damage. Moreover, obese people often consume fewer fruits and vegetables, which are rich in antioxidants that help neutralize excess ROS. Their diets are also typically rich in processed foods, which contain compounds that form covalent adducts with DNA. These factors—oxidative DNA damage, low antioxidant intake, and DNA adduct formation—synergistically promote carcinogenesis.

Beyond DNA damage, such diets also contribute to the accumulation of harmful metabolites, altered gastrointestinal hormones secretion, microbiome dysregulation, and the formation of an immunosuppressive tumor microenvironment [196]. Another major risk factor for PDAC is cigarette smoking. Tobacco smoke compounds induce DNA damage through ROS overproduction and DNA adduct formation. Chronic inflammation is also common among smokers, leading to the release of pro-inflammatory cytokines that further promote cancer cell proliferation and migration.

Pancreatic cancer remains one of the least understood malignancies, and its incidence continues to rise. A significant proportion of PDAC cases are attributable to environmental factors, and nearly 25% of patients have mutations in DDR genes [197]. This evolutionarily conserved pathway plays a central role in maintaining genome stability, as it is responsible for identifying and repairing DNA damage. In response to the recognition of DNA damage, a network of repair pathways is activated, leading to outcomes such as apoptosis, cell cycle arrest, or DNA repair.

In the future, developing a panel of DDR genes that are dysregulated in PDAC could greatly improve early-stage diagnosis and patient prognosis. Additionally, pancreatic cancer is highly resistant to current chemotherapies. Thus, targeting components of the DDR pathway represents a promising avenue to improve patient outcomes.

Several research groups have focused on therapies targeting DNA repair mechanisms. The Know Your Tumor study, conducted in the USA between 2014 and 2019, assessed overall survival in PDAC patients receiving personalized therapies. Molecular profiling revealed that 12–25% of pancreatic tumors contain so-called actionable molecular alterations, defined as a molecular changes for which there is clinical or preclinical evidence of benefit from patient-tailored therapies [197]. A large proportion of these alterations were found in DDR-related genes [197]. In this retrospective study, DDR alterations were defined as mutations in any of the following genes: *BRCA1*, *BRCA2*, *PALB2*, *ATM*, *ATR*, *ATRX*, *BAP1*, *BARD1*, *BRIP1*, *CHEK1*, *CHEK2*, *RAD50*, *RAD51*, *RAD51B*, *FANCA*, *FANCC*, *FANCD2*, *FANCE*, *FANCF*, *FANCG*, or *FANCL* [197]. Personalized therapies were then selected for patients with these mutations.

Out of the 189 patients with tumors containing molecular changes, 46 received matched therapies. These therapies included platinum-based compounds, PARP inhibitors targeting *BRCA1* and *BRCA2* mutations, or checkpoint inhibitors for tumors with changes in the MMR pathway [197]. The median survival among patients receiving matched therapy was extended by 1 year compared to the control group [197]. While this result may not appear groundbreaking at first glance, it represents significant progress in the context of a notoriously difficult to treat cancer, and it underscores the importance of further research into DDR mechanisms.

Ongoing research also focuses on inhibiting PARP, which is the main protein detecting SSBs or DSBs [198]. PARP is involved in the recruiting of XRCC1 (in the BER pathway) and participates in both NER and HR [198]. The POLO phase III clinical trial was designed to test the efficacy of olaparib, a PARP inhibitor, in patients with BRCA-mutated metastatic PDAC who had previously received platinum-based chemotherapy [199]. The results demonstrated significantly longer progression-free survival in the olaparib-treated group compared to the placebo group. Another PARP inhibitor under investigation is Talazoparib, which induces DNA repair markers such as γH2AX at much lower concentrations than earlier-generation PARP inhibitors [200]. Talazoparib was evaluated in phase I trials in patients with one of the following cancers: triple-negative breast cancer, ovarian cancer, prostate cancer, pancreatic cancer, and identified germline mutations of *BRCA1* and *BRCA2*. Of the 13 pancreatic cancer patients enrolled, 4 experienced clinical benefit [200].

Research is also underway to implement therapies targeting other repair pathways, including MMR. Mutations in key MMR genes can lead to MMR deficiency, resulting in the loss of functional repair and microsatellite instability [201]. Interestingly, cancer cells can exploit immune checkpoints to evade immune detection [201]. In this context, therapies using immune checkpoint inhibitors have shown promise. Pembrolizumab, an anti-PD-1 antibody, was tested in phase II clinical trials in patients with PDAC with high microsatellite instability [202]. Previously untreated patients received a combination of pembrolizumab and gemcitabine, and the results showed improved efficacy: partial response, longer median progression-free survival, increased overall survival, and a greater reduction in tumor cell-free DNA copy number instability [202].

Finally, ATR kinase is emerging as another promising therapeutical target. ATR, a PIKK-related kinase, responds to the presence of ssDNA during replication stress, NER, or HR [203]. It phosphorylates multiple substrates involved in DNA repair, controls the activation of replication or the launch of replication forks, and is even responsible for cell cycle arrest [204]. ATR is critical for replication regulation in both normal and cancerous cells. Several ATR inhibitors are currently in clinical trials, including berzosertib, gartisertib, ceralasertib, camonsertib, and elimusertib [205,206,207,208,209]. Some of these, like ceralasertib, are first-line oral ATR inhibitors, while others, such as berzosertib, are being studied in combination with PARP inhibitors or anti-PDL1 antibodies. However, most clinical studies involving these agents have focused on non-small-cell lung cancer or melanoma. There is currently a lack of data on their efficacy in PDAC, highlighting the need for further research, specifically in pancreatic cancer populations.

Although the role of genetic alterations in DDR genes in PDAC is gaining increasing attention, our understanding of their therapeutic potential remains limited. Preclinical studies have demonstrated that *BRCA2*-deficient pancreatic cancer cells are markedly more sensitive to the PARP inhibitor Talazoparib, both in vitro and in mouse xenograft models [210]. This increased sensitivity is consistent with the principle of synthetic lethality, whereby inhibition of PARP in the context of HR deficiency leads to unrepaired DNA damage and tumor cell death [210]. The clinical trials summarized in Table 3 demonstrate some promise, particularly in tumors harboring *BRCA1/2* mutations. For instance, the POLO trial showed that maintenance therapy with olaparib nearly doubled progression-free survival (7.4 vs. 3.8 months) but without a significant improvement in overall survival (19 vs. 19.2 months) in patients with germline *BRCA* mutations [161]. Moreover, studies involving non-BRCA DDR mutations reported only modest efficacy, with median progression-free survival of 3.7 months and overall survival around 9.9 months [190].

## 8. Conclusions

In this review, we have shown that the main risk factors for the development of PDAC are environmental. Smoking, obesity, and type 2 diabetes contribute to the formation of DNA damage, which disrupts genome stability and promotes carcinogenesis. Eliminating these factors may help prevent the development of this cancer.

Pancreatic cancer is a drug-resistant neoplasm often diagnosed at advanced stages and, due to limited treatment options, is associated with a high mortality rate. The mechanisms underlying its aggressiveness are not yet fully understood. A promising new direction in pancreatic cancer research involves targeting DNA repair pathways. DNA damage repair mechanisms are critical defense systems that maintain genomic stability. Defects in these pathways are recognized risk factors for the development of many cancers, including pancreatic cancer.

Therefore, there is an urgent need to better understand the mechanisms linking DDR to the progression and invasiveness of pancreatic cancer. This knowledge may provide a foundation for developing new therapies aimed at improving patient prognosis. Currently, DDR genes are not routinely tested during diagnosis. Although studies on PDAC patients often involve relatively small groups of patients at various stages, the available evidence supports a focus on molecular alterations in DDR and highlights the potential effectiveness of personalized therapies targeting these mechanisms in the treatment of PDAC.

## Figures and Tables

**Figure 1 ijms-26-10106-f001:**
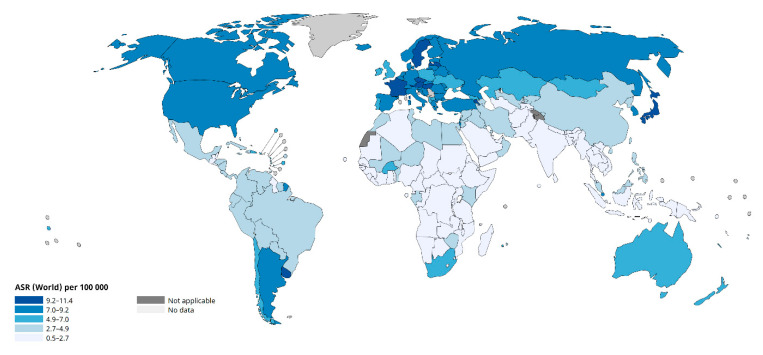
Global ASR (age-standardized rate) per 100,000 cases of pancreatic cancer for both sexes in 2022. Source: WHO (World Health Organization), International Agency for Research on Cancer.

**Figure 2 ijms-26-10106-f002:**
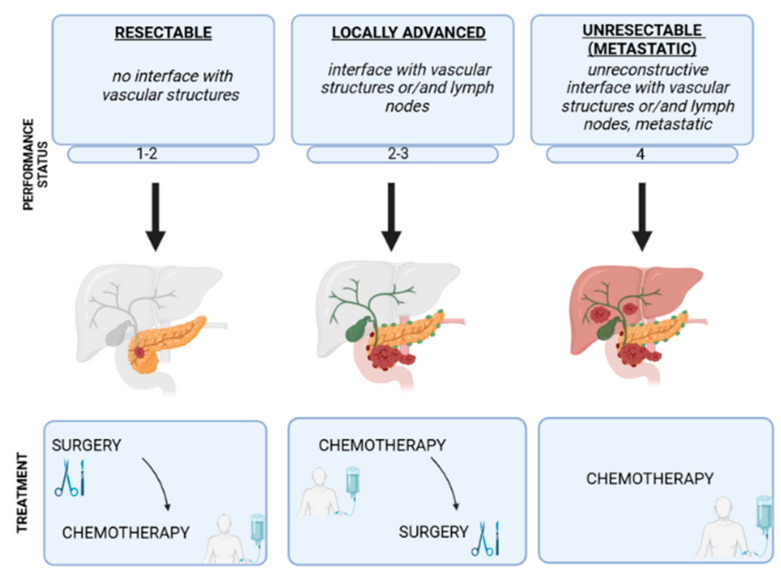
The choice of treatment depends on the stage of the disease. In stages 1–2, the most common method is surgical resection followed by adjuvant chemotherapy. In stages 2–3, the first line of treatment is chemotherapy, followed by partial resection if possible. In stage 4, chemotherapy and palliative care are usually used.

**Figure 3 ijms-26-10106-f003:**
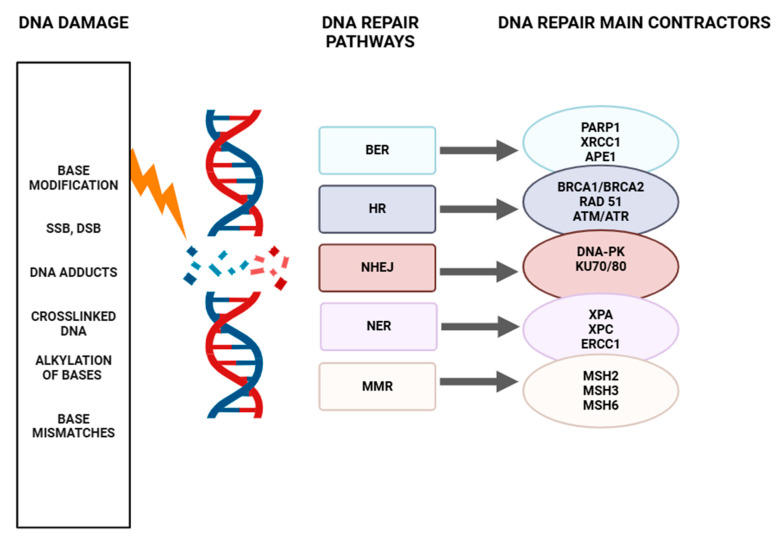
The types of DNA damage and repair pathways. Depending on the DNA damaging agent, different repair pathways are activated. Each pathway has its own characteristic key player. Legend: SSB—single-strand break, DSB—double-strand break, BER—Base Excision Repair, HR—homologous repair, NHEJ—non-homologous end joining, NER—nucleotide excision repair, and MMR—mismatch repair. The abbreviations and the role of the main DNA repair contractors are explained in Section 5.1, Section 5.2, Section 5.3, Section 5.4 and Section 5.5.

**Figure 4 ijms-26-10106-f004:**
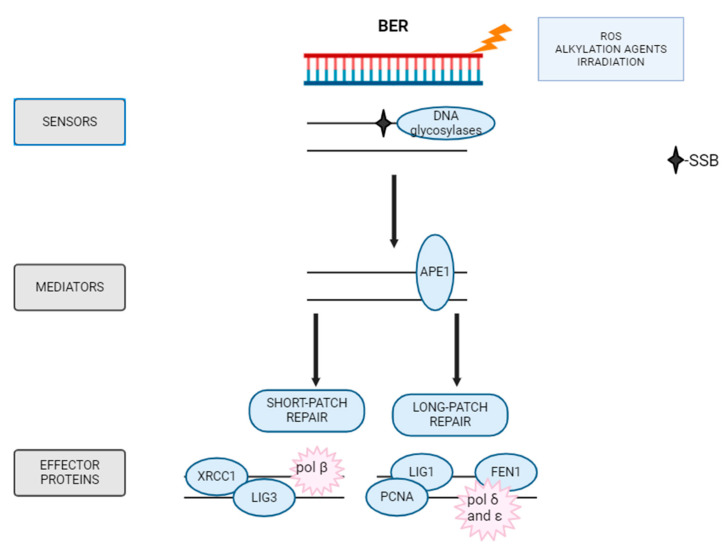
A schematic illustration of the BER pathway for damaged bases and DNA strand breaks. After the recognition of DNA damage by the specific DNA glycosylases, an AP site is created. Then, the APE1 endonuclease cleaves the site. Next, the effector proteins together with the polymerase specific for specific the BER pathway fill the gap and seal the DNA ends.

**Figure 5 ijms-26-10106-f005:**
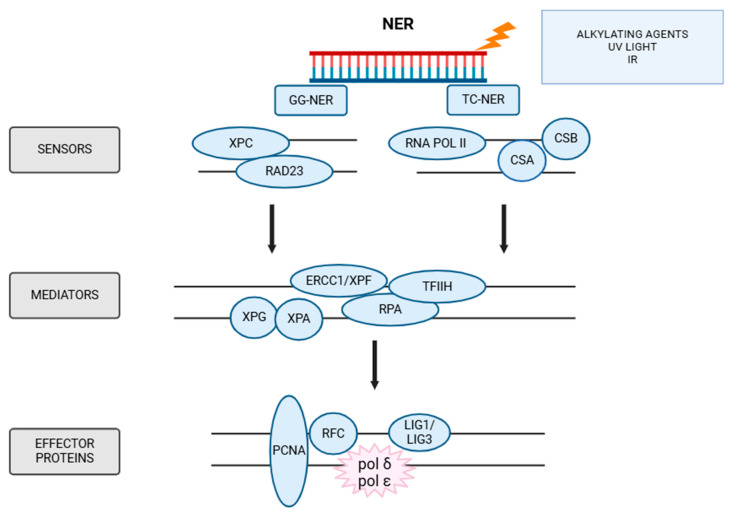
An overview of the NER repair process. This repair pathway is divided into global genomic NER (GG-NER) and transcription-coupled NER (TC-NER). In GG-NER, the damage is recognized by the XPC-RAD23 complex. The TFIIH complex is then recruited, and the XPF-ERCC1 endonuclease excises the damaged DNA. The resulting gap is filled in by DNA polymerases, and the strand is sealed by a ligase. In TC-NER, CSB binds to the stalled RNA polymerase II and recruits CSA. TFIIH is subsequently recruited to the damage site, and together with RPA and the XPF-ERCC1 complex, the damaged DNA strand is excised. DNA polymerases cut the damage strands and polymerases fill the gap.

**Figure 6 ijms-26-10106-f006:**
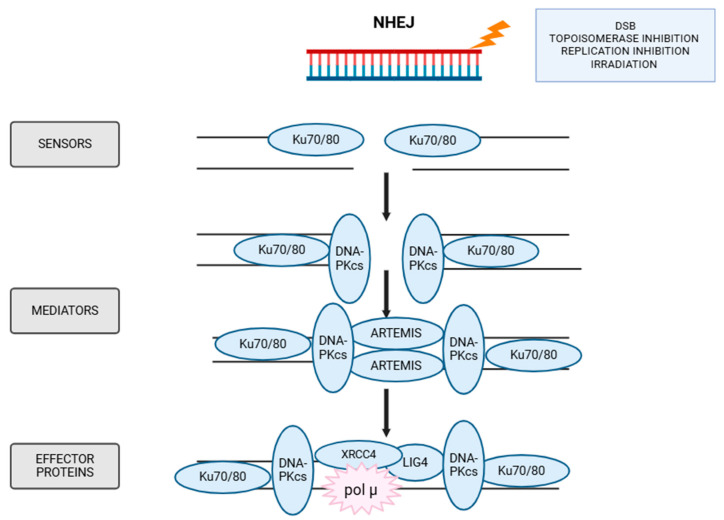
A simplified model of the NHEJ process. This pathway begins with the recognition of DNA double-strand break ends by the Ku70/80 heterodimer, which recruits the DNA-PKcs. Subsequently, the nuclease Artemis processes the DNA ends. Finally, the XRCC4 with Ligase IV complex, together with the specific polymerase, fills in missing nucleotides and seals the break.

**Figure 7 ijms-26-10106-f007:**
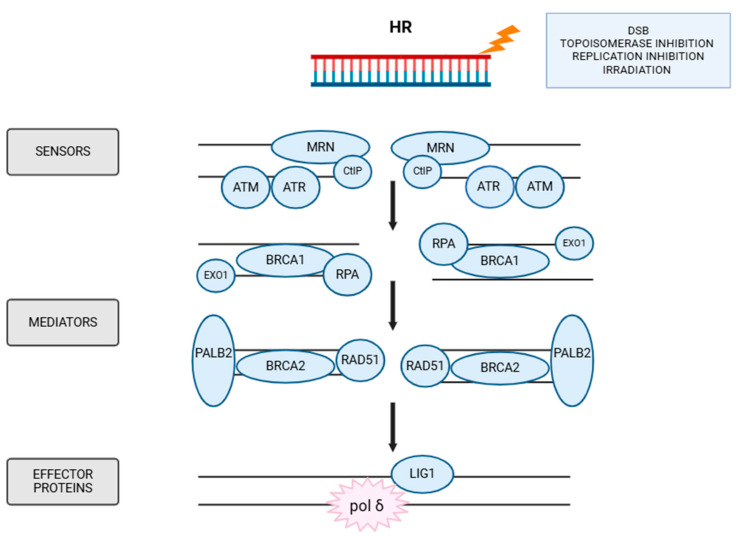
The HR pathway. The damage is recognized by ATM and the MRN complex with CtIP, which initiates DNA end resection. The nucleases and helicases extend the resection, generating ssDNA coated by RPA. BRCA1/BRCA2 mediate the replacement of RPA with RAD51, forming a nucleoprotein filament. RAD51 promotes strand invasion into the sister chromatid, creating a D-loop and enabling DNA synthesis by a polymerase.

**Figure 8 ijms-26-10106-f008:**
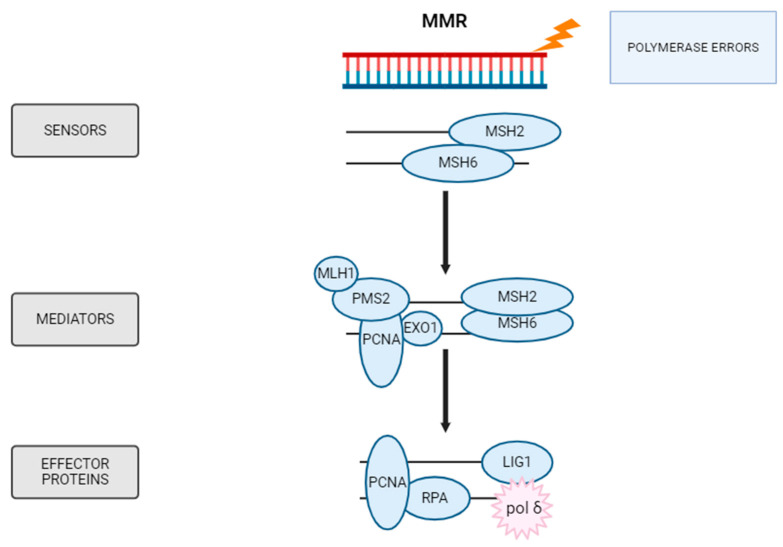
The MMR pathway with the roles of the relevant proteins. After the detection of mismatches, protein complexes such as MutS bind to base mismatches. Then, MLH-1 with PMS-2 is recruited, which activates downstream processes. The exonuclease EXO1 excises the DNA fragment containing the mismatch. Finally, DNA polymerase fills in the gap with the correct nucleotides, and a DNA ligase seals the strand.

**Table 1 ijms-26-10106-t001:** The prevalence of pancreatic cancer among different populations in both sexes in 2022. Source: WHO, International Agency for Research on Cancer.

Population	Number	ASR ^1^ World
Africa	18,993	2.4
Asia	232,537	3.6
** Europe **	** 146,477 **	** 8.0 **
Latin America and the Caribbean	41,032	4.6
** Northern America **	** 67,089 **	** 8.5 **
Oceania	4864	6.2
Total	510,992	4.7

^1^ ASR: age-standardized rate.

**Table 2 ijms-26-10106-t002:** A summary of types of DNA damage and their sources.

Type of DNA Damage	Alkylation of Bases	Purine and Pyrimidine Dimers	DNA Adducts	ICL ^1^ and DPC ^2^	Oxidative DNA Damage	DNA Strand Breaks
Source	Endogenous processes (byproducts of oxidative damage, cellular methylation donors) Environmental compounds (tobacco smoke)	UVR ^3^, IR ^4^ (sunlight, X-rays)	Binding with a chemical substance (tobacco smoke, alcohol metabolism, processed food)	Byproduct of cell metabolism Exogenous factors (IR, UV, chemotherapeutics)	Oxidative stress (exposure to increased or decreased oxygen pressure, the action of chemical compounds, radiation, inflammation)	Endogenous DNA metabolic processes (activity of DNA topoisomerase, impaired DNA replication, transcription, and recombination) IR Anticancer drugs (cisplatin)

Legend: ^1^ ICL—interstrand crosslink, ^2^ DPC—DNA–protein cross-link, ^3^ UVR—ultraviolet radiation, and ^4^ IR—ionizing radiation.

**Table 3 ijms-26-10106-t003:** Summary of clinical trials focused on DDR in PDAC.

Trial	Design	Patients	Treatment	References
NCT02950064	Phase I, open label	Patients with *BRCA1/BRCA2* or other DNA repair mutations with advanced solid tumors (one of the following cancers: **pancreatic cancer**, castration-resistant prostate cancer, ovarian cancer, triple-negative breast cancer)	BTP-114, novel platinum compound	-
NCT01489865	Phase I/II study, single arm	Patients with metastatic PDAC with *BRCA/PALB2/FANC* mutations or family history	ABT-888 (PARP inhibitor) combined with mFOLFOX6 (oxaliplatin, 5-FU/leucovorin)	[188]
NCT02184195	Phase III, randomized	Metastatic PDAC with germline *BRCA1/BRCA2*, no progression on first-line platinum-based treatment	Olaparib or placebo treatment	[161]
NCT01585805	Phase II, randomized	Locally advanced or metastatic PDAC with *BRCA1* or *PALB2* mutations	Veliparib, gemcitabine, cisplatin (Arm A); gemcitabine, cisplatin (Arm B); placebo (Arm C)	[189]
NCT02042378	Phase II, single arm	PDAC with *BRCA* mutation	Rucaparib (PARP inhibitor) treatment	[190]
NCT02184195	Phase II, randomized	PDAC patients with documented mutations in *BRCA1/BRCA2*	Olaparib vs. placebo treatment	[191]
NCT03140670	Phase II, open-label	Platinum-sensitive advanced PDAC with *BRCA1/BRCA2* or *PALB2* mutations	Rucaparib (PARP inhibitor) treatment	[192]
NCT03682289	Phase II	Advanced solid tumors (**including PDAC**) with progression	Ceralasertib (ATR Kinase Inhibitor) alone (Arm A), ceralasertib; olaparib, (Arm B), ceralasertib, Durvalumab (Arm C)	-
NCT03669601	Phase I, non-randomized (dose escalation)	Inoperable/unresectable locally advanced or metastatic PDAC and other solid tumors	AZD6738 (ATR inhibitor)	-
NCT03404960	Phase Ib/II open label	PDAC patients who received prior platinum-based treatment	Niraparib + Nivolumab (Arm A); Niraparib + Iplimumab (Arm B)	[193]

## Data Availability

No new data were created or analyzed in this study.

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
