# Peer review of "DNA Damage and Repair in Pancreatic Cancer—The Latest Findings"

_ijms, 2025, doi:10.3390/ijms262010106_

Round 1

Reviewer 1 Report

Comments and Suggestions for Authors

The article reviewed the role of DNA Damage and Repair pathways in PDAC. Importantly, the authors highlighted the utility of these pathways as therapeutic targets against this aggressive and lethal cancer. This is a relevant and adequately reviewed article. Please see some comments to address:

  1. Please ensure that statements are well-referenced. Some examples that require references are Line 78-79, Line 109-110, Line 139-140
  2. Line 140- What factors are the authors referring to
  3. Line 151- The Authors say the incidences are “negligible”, please correct this with appropriate wording.
  4. Line 186-187- Please rewrite for clarity
  5. In general, authors should ensure that figure legends and titles are appropriately descriptive
  6. Also, ensure that abbreviations used in tables and figures are defined in the legends or as footnotes for tables
  7. Page 7- The staging and treatment for PDAC can be summarised adequately.
  8. The authors should comment on the utility of targeting multiple DDR-related genes for therapy.

Author Response

We would like to thank the Reviewers for insightful comments on our manuscript. We appreciate the time and effort that was dedicated to provide the feedback on our paper, and we are grateful for valuable suggestions. We have now responded point-by-point to the comments and concerns that were raised. We feel that the revised manuscript is clearer.

Point 1: Please ensure that statements are well-referenced. Some examples that require references are Line 78-79, Line 109-110, Line 139-140

Answer: Thank you for this comment. The references have been added.

Point 2: Line 140- What factors are the authors referring to.  

Answer: Thank you for this remark. The sentence has been clarified:  The lifestyle and environmental factors such as cigarette smoking, alcohol consumption, processed food intake or environmental toxins are a source of molecules that induce a series of changes in DNA, leading to carcinogenesis [25].

Point 3: Line 151- The Authors say the incidences are “negligible”, please correct this with appropriate wording

Answer: Indeed, we made a mistake. The word „negligible” has been replaced with „insignificant”.

Point 4: Line 186-187- Please rewrite for clarity

Answer: Thank you for this remark. The sentence has been modified:  Risk factors for pancreatic cancer include pancreatitis, which can result from excessive alcohol consumption or have hereditary basis.

Point 5: In general, authors should ensure that figure legends and titles are appropriately descriptive

Answer: Thank you for your valuable comment. The descriptions of tables and figures have been changed.

Table 1: The prevalence of pancreatic cancer among different populations in both sexes in 2022.
Figure 2: The choice of treatment depends on the stage of the disease. In stages 1-2, the most common method is surgical resection followed by adjuvant chemotherapy. In stages 2-3, the first line of treatment is chemotherapy, followed by partial resection if possible. In stage 4, chemotherapy and palliative care are usually used.

Figure 3: The types of DNA damage and repair pathways. Depending on the DNA-damaging agent, different pathways are activated. Each pathway has its own characteristic key player.  

Figure 4: A schematic illustration of the BER pathway for damaged bases and DNA strand breaks. After the recognition of DNA damage by the specific DNA glycosylases, an AP site is created. Then the APE1 endonuclease cleaves the site. Next, the effector proteins together with the polymerase specific for specific BER-pathway fill the gap and seal the DNA ends.  

Figure 5: An overview of the NER repair process. This repair pathway is divided into global genomic NER (GG-NER) and transcription-coupled NER (TC-NER). In GG-NER, the damage is recognized by the XPC-RAD23 complex. The TFIIH complex is then recruited, and the XPF-ERCC1 endonuclease excises the damaged DNA. The resulting gap is filled in by DNA polymerases, and the strand is sealed by ligase. In the TC-NER, CSB binds to the stalled RNA polymerase II and recruits CSA. TFIIH is subsequently recruited to the damage site, and together with RPA and the XPF-ERCC1 complex, the damaged DNA strand is excised. DNA polymerases cut the damage strands and polymerases fill the gap.

Figure 6: A simplified model of the NHEJ process. This pathway begins with the recognition of DNA double-strand break ends by the Ku70/80 heterodimer, which recruits the DNA-PKcs. Subsequently, the nuclease Artemis processes the DNA ends. Finally, the XRCC4 with Ligase IV complex, together with the specific polymerase, fills in missing nucleotides and seals the break.

Figure 7: The HR pathway. The damage is recognized by ATM and MRN complex with CtIP, which initiates the DNA end resection. The nucleases and helicases extend the resection, generating ssDNA coated by RPA. BRCA1/BRCA2 mediate replacement of RPA with RAD51, forming a nucleoprotein filament. RAD51 promotes strand invasion into the sister chromatid, creating a D-loop and enabling DNA synthesis by polymerase.

Figure 8: The MMR pathway with the roles of the relevant proteins. After detection of mismatches, protein complexes such as MutS bind to base mismatches. Then MLH-1 with PMS-2 is recruited, which activates downstream processes. The exonuclease EXO1 excises the DNA fragment containing the mismatch. Finally, DNA polymerase fill the gap with the correct nucleotides, and DNA ligase seals the strand.

Point 6: Also, ensure that abbreviations used in tables and figures are defined in the legends or as footnotes for tables

Answer: Thank you for this valuable suggestion. The abbreviations have been explained.
Table 2: Legend: 1 ICL- interstrand crosslinks, 2 DPC- DNA-protein cross-links, 3 UVR- Ultraviolet radiation, 4 IR- Ionizing Radiation.

Figure 3: The types of DNA damage and repair pathways. Legend: SSB- single-strand breaks, DSB- double-strand breaks, BER- Base Excision Repair, HR- Homologous Repair, NHEJ- non-homologous end joining, NER- nucleotide excision repair, MMR- mismatch repair. The abbreviation and role of DNA repair main contractors are explained in the subsection 5.1-5.5. 

Point 7: Page 7- The staging and treatment for PDAC can be summarised adequately.

Answer: Thank you for this prompt remark. A brief summary has been added: Taken together, the diagnostic options for pancreatic cancer remain limited and rely primarily on imaging techniques, such as computed tomography and magnetic resonance imaging techniques. To date, no specific biomarkers have been identified that would allow for early-stage detection of the disease. Therapeutic strategies are likewise constrained, with treatment choices largely determined by the diagnosis stage. Among the available approaches, surgical resection combined with chemotherapy is recommended in early stages, while neoadjuvant chemotherapy and, where feasible, partial resection may be considered.

Point 8: The authors should comment on the utility of targeting multiple DDR-related genes for therapy.

Answer: Thank you this  comment. We added our comments on the utility oftargeting multiple DDR-related genes for therapy.

Although the role of  genetic alterations in DDR genes in PDAC is gaining increasing attention, our understanding of their therapeutic potential remains limited. Preclinical studies have demonstrated that BRCA2 deficient pancreatic cancer cells are markedly more sensitive to the PARP inhibitor- talazoparib, both in vitro and in mouse xenograft models [212]. This increased sensitivity in consistent with the principle of synthetic lethality, whereby inhibition of PARP in the context of HR deficiency leads to unrepaired DNA damage and tumor cell death [212].  Clinical trials summarized in Table 3 demonstrate some promise, particularly in tumors harboring BRCA1/2 mutations. For instance, the POLO trial showed that maintenance therapy with olaparib  nearly doubled progression-free survival (7.4 vs 3.8 months) but without a significant improvement in overall survival (19 vs 19.2 months) in patients with germline BRCA mutations [162]. Moreover, studies involving non-BRCA DDR mutations reported only modest efficacy, with median progression-free survival of 3.7 months and overall survival around 9.9 months [192]. These findings highlight the ongoing need to identify additional DDR-related targets and combination strategies to enhance treatment outcomes. Targeting multiple DDR-related pathways simultaneously could help overcome compensatory repair mechanisms and increase tumor vulnerability to genotoxic agents, offering a future direction to improve survival in this highly aggressive cancer type.

Reviewer 2 Report

Comments and Suggestions for Authors

In this review article, the authors summarize the incidence, etiology, prognosis, and other aspects of pancreatic cancer. They also discuss the types of DNA damage and provide an overview of DNA damage repair (DDR) pathways. While the sections on DNA damage and DDR pathways are detailed and comprehensive, they remain somewhat general and lack pancreatic cancer–specific examples, which should be the primary focus of this review given its title. The authors should include and discuss studies that report mutations in DDR pathway genes in PDAC to better highlight the dynamics of DDR in this context. In addition, several ongoing clinical trials are investigating the role of DDR in PDAC; a summary of these trials should also be incorporated to strengthen the review.

Author Response

We would like to thank the Reviewers for insightful comments on our manuscript. We appreciate the time and effort that was dedicated to provide the feedback on our paper, and we are grateful for valuable suggestions. We have now responded point-by-point to the comments and concerns that were raised. We feel that the revised manuscript is clearer.

Point: In this review article, the authors summarize the incidence, etiology, prognosis, and other aspects of pancreatic cancer. They also discuss the types of DNA damage and provide an overview of DNA damage repair (DDR) pathways. While the sections on DNA damage and DDR pathways are detailed and comprehensive, they remain somewhat general and lack pancreatic cancer–specific examples, which should be the primary focus of this review given its title. The authors should include and discuss studies that report mutations in DDR pathway genes in PDAC to better highlight the dynamics of DDR in this context. In addition, several ongoing clinical trials are investigating the role of DDR in PDAC; a summary of these trials should also be incorporated to strengthen the review.

Answer: Thank you for this valuable comment. We searched the literature for additional data concerning DNA damage and repair specifically to PDAC. The manuscript has been enriched with the following paragraphs addressing to these issues:

(lines 375-396) Li et al. conducted a study to investigate the role of carcinogen exposure in PDAC [92]. They measured total aromatic and lipid-peroxidation related DNA adducts in pancreatic tumor tissues and adjacent noncancerous pancreatic tissues obtained from patients undergoing surgical resection [92]. The study demonstrated that tumor tissues contained significantly higher levels of DNA adducts than adjacent normal tissues. Although specific chemical identities of the adducts were not determined, these DNA modifications are likely derived from PAH from cigarette smoke and HCA (heterocyclic amines) from diet, as well as from lipid peroxidation products such as MDA (malondialdehyde) [92]. Additionally, Li et al. correlated DNA adduct levels with KRAS mutations and genetic polymorphisms in DNA repair genes, suggesting a link between environmental exposure, DNA damage and pancreatic carcinogenesis   [92]. Similarly, Wang et al. investigated the potential impact of environmental and dietary carcinogens on pancreatic carcinogenesis [93]. Specifically, they examined DNA adducts formed by PAHs derived from cigarette smoke and lipid peroxidation products, mainly MDA , which can result from oxidative degradation of polynunsaturated fatty acids [93]. Using the ^32P-postlabeling method, the authors quantified aromatic and lipid peroxidation-related DNA adducts in pancreatic tumor tissues and adjacent noncancerous tissues obtained from patients with pancreatic cancer [93]. They observed significantly higher levels of MDA-DNA adducts in cancerous tissues compared with adjacent normal tissues [93]. These findings suggest that DNA damage resulting drom both PAH-derived and lipid peroxidation derived adduct formation may contribute to pancreatic carcinogenesis  [93].

(lines 436-439)Mohamadkhani et al. demonstrated that patients with pancreatic cancer had significantly higher levels of 8-OHdG (8-hydroxy-2’-deoxyguanosine) in peripheral blood leukocytes, a marker of oxidative DNA damage, compared to healthy individuals [110].

(lines 463-476)  Li YH et al. conducted in vitro studies on 18 PDAC cell lines to evaluate the role of DNA repair in PDAC. They observed that PDAC cells are characterized by elevated  levels of γH2AX foci, which is a marker of DNA damage, primarily DSB [119]. They also showed that inhibition of the NHEJ repair pathway by inhibiting DNA-Pkcs, a key subunit of the NHEJ pathway, further increases DSB accumulation [119]. Next, the response of PDAC cells to DNA damage afterIR was examined at different time points by measuring γH2AX foci [119]. It was found that DNA repair was significantly impaired by DNA-Pkcs inhibition [119]. Importantly, NHEJ inhibition sensitizes PDAC cells to IR, which was verified by DNA-Pk inhibition followed by irradiation at doses of 2, 4 and 6 Gy (grey) [119]. This findings highlight a potential therapeutic window for exploring synthetic lethality. If PDAC cells already exhibit HR defects, additional inhibition of NHEJ may result in the accumulation of unrepaired DSBs, ultimately triggering cancer cell death. Therefore, comprehensive profiling of DDR in tumors may inform personalized treatment strategies targeting synthetic lethality. 

(lines 675-688) It is important to emphasize that alterations in these pathways, especially mutations in DDR genes, have significant implications for PDAC development and progression. Understanding the landscape of DDR gene mutations provides insight into the molecular mechanisms driving genomic instability in PDAC, as well as potential therapeutic strategies. Both germline and somatic mutations have been identified in key DDR genes such as BRCA1/BRCA2, PALB2, RAD51, FANCB, ARID1A and ATM [161-163]. Dysfunction of these geneses and their encoded proteins leads to defects in repairing DSBs, particularly through HR deficiency [164]. In PDAC such defects contribute to genomic instability, facilitating the accumulation of mutations and tumor progression [165] . For example, mutations in BRCA2 and PALB2 impair the ability to accurately repair DSBs, affecting tumor cell proliferation and survival [165]. Additionally, mutations in ATM and RAD51 disrupt DNA damage signaling and checkpoint control, which may lead to compromised cell cycle regulation and defective apoptosis [166].

(lines 764-812)The role of DDR in PDAC is being studied not only in in vitro models, but also in animals. Rowley et al. developed a mouse model with PDAC-specific KRAS and TP53 mutations with BRCA2 inactivation [185]. Histological analysis of mouse pancreatic tissues revealed that BRCA2 inactivation promoted the development of precancerous lesions and pancreatic tumors [185]. In addition, cancer cells were isolated from mice and then tested for sensitivity to chemotherapeutic agents, demonstrating that BRCA2-deficient pancreatic tumors exhibit increased sensitivity to cisplatin and PARP inhibitors [185]. Similarly, Drosos et al. investigated how ATM gene deletion affects a mouse model of pancreatic cancer with oncogenic KRAS expression [186]. Their research showed that partial or complete ATM deficiency interacts with the KRAS gene to promote highly metastatic pancreatic cancer and also leads to permanent DNA damage in precancerous lesions and primary tumors [186]. Other studies investigating ATM deficiency in a mouse model of PDAC were conducted by Perkhofer et al. [187]. In this study ATM-deficient mice were subcutaneously implanted with PDAC cells to induce tumor formation. Once tumors had developed, the mice were treated with gemcitabine, olaparib or a combination of both drugs [187]. Analysis of the isolated tumors revealed that loss of ATM accelerated PDAC progression and was associated with enhanced cancer cell stemness and EMT [187]. Furthermore, ATM deletion impaired the DDR, as evidenced by a sixfold increase in γH2AX and 53BP1 foci compared to control tumors [187]. Since ATM deficiency compromises HR repair, the authors further explored the effect of PARP inhibition in this context. PDAC cells isolated from ATM-deficient tumors were treated in vitro with olaparib and gemcitabine in a clonogenic survival assay [187]. This treatment resulted in significantly reduced colony formation compared to controls  [187].

DDR signaling in PDAC is currently under investigation in clinical trials. DDR-deficient PDAC subtypes, particularly those harboring BRCA1/BRCA2 mutations, have shown sensitivity to platinum-based chemotherapy and to PARP inhibitions, such as olaparib [188]. In a non-randomized phase II  clinical trial, patients with advanced PDAC carrying mutations in at least one of the following DDR genes: ATM, RAD51, ARID1A, PALB2, FANCB or BRCA1/2 were treated with olaparib [188]. The study demonstrated that these patients had a significantly longer median progression-free survival compared to historical controls, especially those who had shown prior sensitivity to platinum agents [188]. In another clinical study, the therapeutic relevance of DDR deficiency was further explored through retrospective analysis of metastatic PDAC patients treated with FOLFIRINOX [189]. Sehdev et al. observed that patients with germline or somatic mutations in DDR genes exhibited significantly improved overall survival compared to non-carries [189]. These findings support the prognostic and potentially predictive value of DDR mutations in selecting treatment regimens involving DNA-damaging agents [189].  

These findings highlight the ongoing need to identify additional DDR-related targets and combination strategies to enhance treatment outcomes. Targeting multiple DDR-related pathways could help overcome compensatory repair mechanisms and increase tumor vulnerability to genotoxic agents, offering a future direction to improve survival in this highly aggressive cancer type. Two complementary strategies have emerged for exploiting synthetic lethality in PDAC, targeting HR-deficient tumors with PARP inhibitors and sensitizing HR-proficient tumors by inhibiting NHEJ. Comprehensive analysis of DDR related mutations and functional defects is thus essential to guide personalized therapy, maximize DNA damage accumulation and promote selective tumor cell death. A summary of clinical trials related to DDR in PDAC is provided in the Table 3.

Table 3. Summary of clinical trials focused on DDR in PDAC.

Trial

Design

Patients

Treatment

References

NCT02950064

Phase I, open-label

Patients with  BRCA1/BRCA2 or other DNA-repair mutations with advanced solid tumors (one of the following cancers: pancreatic cancer, castration-resistant prostate cancer, ovarian cancer, triple-negative breast cancer)

BTP-114, novel platinum compound

-

NCT01489865

Phase I/II study, single arm

Patients with metastatic PDAC with BRCA/PALB2/FANC mutations or family history

ABT-888 (PARP inhibitor) combined with mFOLFOX6 (oxaliplatin, 5-FU/leucovorin)

[190]

NCT02184195

Phase III, randomised

Metastatic PDAC with germline BRCA1/BRCA2, no progression on first-line platinum-based treatment

Olaparib or placebo treatment

[162]

NCT01585805

Phase II, randomized

Locally advanced or metastatic PDAC with BRCA1 or PALB2 mutations

veliparib, gemcitabine, cisplatin (Arm A),
gemcitabine, cisplatin (Arm B), placebo (Arm C)

[191]

NCT02042378

Phase II, single-arm

PDAC with BRCA mutation

Rucaparib ( PARP inhibitor) treatment

[192]

NCT02184195

Phase II, randomized

PDAC patients with documented mutations in BRCA1/BRCA2

Olaparib vs placebo treatment

[193]

NCT03140670

Phase II, open-label

Platinum-sensitive advanced PDAC with BRCA1/BRCA2 or PALB2 mutations

Rucaparib (PARP inhibitor) treatment

[194]

NCT03682289

Phase II

Advanced solid tumors (including PDAC) with progression

ceralasertib (ATR Kinase Inhibitor) alone (Arm A), ceralasertib, olaparib, (Arm B),  Ceralasertib, Durvalumab (Arm C)

-

NCT03669601

Phase I, non-randomised (dose escalation)

Inoperable/unresectable locally advanced or metastatic PDAC and other solid tumors

AZD6738 (ATR inhibitor)

-

NCT03404960

Phase Ib/II open-label

PDAC patients who received prior platinum based treatment

Niraparib+Nivolumab (Arm A),

Niraparib+ Iplimumab (Arm B)

[195]

Round 2

Reviewer 2 Report

Comments and Suggestions for Authors

All reviewer concerns have been addressed satisfactorily by the authors. I recommend the manuscript for acceptance.